# Pilot Testing of a Patient Decision Aid for Adolescents with Severe Obesity in US Pediatric Weight Management Programs within the COMPASS Network

**DOI:** 10.3390/ijerph16101776

**Published:** 2019-05-20

**Authors:** Jaime Moore, Matthew Haemer, Nazrat Mirza, Ying Z Weatherall, Joan Han, Caren Mangarelli, Mary Jane Hawkins, Stavra Xanthakos, Robert Siegel

**Affiliations:** 1Department of Pediatrics, Section of Nutrition, University of Colorado School of Medicine, Aurora, CO 80045, USA; matthew.haemer@ucdenver.edu; 2Children’s National Medical Center, Washington, DC 20010, USA; nmirza@childrensnational.org; 3Department of Surgery, University of Tennessee Health Science Center, Memphis, TN 38103, USA; yzhuge@uthsc.edu; 4Department of Pediatrics, University of Tennessee Health Science Center, Memphis, TN 38103, USA; jhan14@uthsc.edu; 5Department of Physiology, University of Tennessee Health Science Center, Memphis, TN 38103, USA; 6Children’s Foundation Research Institute, Le Bonheur Children’s Hospital, Memphis, TN 38103, USA; 7Duke Children’s Healthy Lifestyles Program, Durham, NC 27705, USA; caren.mangarelli@duke.edu; 8Children’s Hospital & Medical Center, Omaha, NE 68114, USA; mhawkins@childrensomaha.org; 9Division of Pediatric Gastroenterology, Hepatology and Nutrition, Cincinnati Children’s Hospital Medical Center, Cincinnati, OH 45267, USA; stavra.xanthakos@cchmc.org; 10Department of Pediatrics, University of Cincinnati College of Medicine, Cincinnati, OH 45267, USA; 11Heart Institute, Cincinnati Children’s Hospital Medical Center, Cincinnati, OH 45267, USA; bob.siegel@cchmc.org; 12Department of Pediatrics, University of Cincinnati College of Medicine, Cincinnati, OH 45267, USA

**Keywords:** shared decision-making, Patient decision aid, adolescent, severe obesity, treatment, lifestyle, bariatric surgery

## Abstract

Shared decision-making (SDM) is a best practice for delivering high-quality, patient-centered care when there are multiple options from which to choose. A patient decision aid (PDA) to promote SDM for the treatment of adolescent severe obesity was piloted among 12–17-year-olds (*n* = 31) from six pediatric weight management programs within the Childhood Obesity Multi Program Analysis and Study System (COMPASS). Medical providers used a brochure that described indications, risks, and benefits of intensive lifestyle management alone versus bariatric surgery plus lifestyle. Immediately after, patients/families completed a survey. Patient/family perceptions of provider effort to promote understanding of health issues, to listen to what mattered most to them, and to include what mattered most to them in choosing next steps averaged 8.6, 8.8, and 8.7, respectively (0 = no effort, 9 = every effort). Nearly all (96%) reported knowing the risks/benefits of each treatment option and feeling clear about which risks/benefits mattered most to them. Most (93%) reported having enough support/advice to make a choice, and 89% felt sure about what the best choice was. Providers largely found the PDA to be feasible and acceptable. This pilot will guide a more rigorous study to determine the PDA’s effectiveness to support decision-making for adolescent severe obesity treatment.

## 1. Introduction

Severe obesity affects an estimated 8.5% of 12–19-year-olds in the United State (US), and the prevalence continues to rise [1]. Severe obesity in childhood, defined as a body mass index (BMI) that is ≥120% of the 95th percentile for age and sex or a BMI ≥35 kg/m^2^ (whichever is lower), is associated with significantly increased risks of cardiometabolic disease, dramatically reduced quality of life, and premature mortality [2,3,4,5]. Treatment of severe obesity often begins with lifestyle interventions targeting nutrition and physical activity behaviors. However, because of the modest positive effects of these interventions on weight status [6], adjunctive treatment options include medically supervised diets (e.g., meal replacement or high-protein/very-low-carbohydrate diet), anti-obesity pharmacotherapy, and bariatric surgery. Bariatric surgery is the most effective and durable treatment for severe obesity [7,8,9,10]. However, among the estimated 3.9 million US adolescents with severe obesity who may meet medical criteria for bariatric surgery [1,11,12], only a small fraction actually undergo the procedure [13,14]. How families approach decision-making for the treatment of an adolescent’s severe obesity, and their confidence and satisfaction with this process are not well established [15]. Shared decision-making (SDM) is a best practice for delivering high-quality, patient-centered care when there are multiple appropriate options from which to choose (Figure 1) [16,17]. Patient decision aids (PDA) can facilitate SDM and improve patient/family knowledge, understanding of risks, and consistency between personal values and healthcare choices [18]. Unlike other health education materials, a decision aid explicitly focuses on a choice that needs to be made, describes the options, and guides individuals to make that choice by comparing evidence-based risks/benefits to their personal values [19].

Thus, the primary aim of this study was (1) to establish whether a PDA for the treatment of adolescent severe obesity was feasible and acceptable to implement within usual pediatric weight management care. Secondary aims were (2) to determine whether use of the PDA could promote core SDM principles at a single time point as perceived by adolescents with severe obesity and their families, and (3) to further refine the PDA using data from patient/family surveys and providers. We hypothesized that providers at 100% of participating sites would be able to incorporate the PDA into their clinical visits, that family survey data would reflect an experience inclusive of multiple dimensions of SDM, and that family and provider feedback would guide further enhancements of the PDA to improve its usability and acceptability.

## 2. Materials and Methods

### 2.1. Study Design and Procedures

The Childhood Obesity Multi Program Analysis and Study System (COMPASS) is a practice-based research network comprising 25 pediatric weight management programs across 14 US states. All COMPASS programs offering both lifestyle and bariatric surgery treatment options were invited to join the study. Twelve sites initially expressed interest, and six programs ultimately participated: Children’s Hospital Colorado Lifestyle Medicine Program, Cincinnati Children’s Hospital Center for Better Health and Nutrition, Duke Healthy Lifestyles Program, Children’s National Washington, District of Columbia (D.C.) IDEAL Clinic, University of Tennessee Health Science Center (Memphis)/Le Bonheur Children’s Hospital Healthy Lifestyle Clinic, and Children’s Hospital and Medical Center Omaha HEROES Pediatric Weight Management Program. All participating sites are stage 4 tertiary care centers as defined by the 2007 Expert Committee framework [21], and offer comprehensive multidisciplinary pediatric weight management services including bariatric surgery.

This study used a common protocol to help standardize delivery of the PDA across the six sites. To improve fidelity of protocol implementation, the COMPASS group trained participating providers through monthly phone calls prior to and throughout the data collection period. Topics included inclusion criteria, recruitment procedures, SDM content and delivery, and data collection.

Recruitment of patients/families took place between December 2016 and September 2017 during routine clinical weight management visits. Each site had the goal of recruiting a convenience sample of five families. Eligible patients were 12–17 years of age, English-speaking, with severe obesity (defined as BMI ≥120% of the 95th percentile for age and sex or ≥35 kg/m^2^, whichever was lower), and met medical criteria for bariatric surgery. There were no incentives offered to patients/families or providers.

The medical provider presented the PDA to the patient/family during a routine clinical encounter. Core elements of a routine visit across sites included a clinical interview (review of the medical/family/social history and medications, exploration of possible genetic/metabolic factors, and evaluation of lifestyle targets: nutrition, physical activity, and sedentary behaviors), objective assessment of weight status/trajectory using BMI, physical exam, screening for obesity comorbidities with labs and other diagnostic tests (e.g., sleep study), and treatment plan. The PDA was introduced to facilitate the treatment component of the discussion, and providers led the families through each section of the decision aid. Providers had discretion as to when during the visit to introduce the PDA, which varied based on practical considerations, including whether it was a new or returning patient, and whether intensive lifestyle and bariatric surgery was previously discussed. The medical provider then answered any questions that arose after reviewing the PDA. Before the end of the visit, the patient and parent/guardian completed an anonymous 13-question survey together (Appendix A).

Parental consent and adolescent assent were assessed by the first two questions of the survey. Some sites additionally required a short “postcard consent” to be presented at the time of recruitment. This study was conducted in accordance with the Declaration of Helsinki. Each site received local Institutional Review Board approval before beginning data collection. All survey data were centrally managed and stored at the Cincinnati Children’s site.

After all patient/family data were collected, qualitative feedback was elicited from the medical provider who primarily used the PDA at each site, with a seven-item questionnaire, conducted via email (Appendix A).

### 2.2. Measures

#### 2.2.1. Patient Decision Aid 

The PDA (Figure 2) was developed through an SDM collaborative initiated by the Cincinnati Children’s James M. Anderson Center for Health Systems Excellence in 2014 [20], and followed key recommendations for the development of PDAs established by the International Patient Decision Aid Standards Collaboration (IPDAS) [22,23]. The collaborative included providers (R.S., S.X.) and patient and family partners from the comprehensive medical and surgical weight management program at Cincinnati Children’s, and was facilitated by faculty and staff from the Anderson Center who are experienced in design and implementation of PDAs [20,24,25]. An initial option grid (a summary table that compares treatments) was developed using available evidence on weight management treatment options and outcomes, and with survey input from providers and patients [26]. The decision aid prototype was then developed and field-tested iteratively in the weight management program at Cincinnati Children’s from October 2014 to March 2015, incorporating feedback from both providers and patients before finalizing the PDA in April 2015. The aid was designed to support decision-making between medical providers and adolescents/families about two major treatment options for severe obesity: intensive lifestyle management and bariatric surgery plus lifestyle.

For this study, other adjunctive treatment options (e.g., medically supervised diets, anti-obesity medications) were not included in the PDA because they (a) were not uniformly offered by every program and (b) are not universally adopted as standard of care [27,28]. Expertise and resources to deliver specialized diets varied by site, and state-specific medical boards differ in their regulation of anti-obesity medications. For example, in Ohio, where the PDA was developed, the state medical board prohibits off-label use of the anorexigenic agent phentermine (including in adolescents under age 17) [29]. In other programs, phentermine is commonly prescribed [30]. In programs where treatment options other than intensive lifestyle and bariatric surgery were offered, the site PI had discretion to modify the PDA to incorporate alternative therapies. One of the six participating sites chose to make this type of modification, and added medications and special diets under the intensive lifestyle section of the PDA. Others verbally discussed available treatment options that did not appear on the PDA. This flexibility was deemed reasonable given the primary study aim of feasibility and acceptability, and helped to ensure that the full range of treatment options was presented to families.

The PDA was written at an eighth-grade reading level and included overviews of the health problem, the treatment decision to be made, and the two major treatment options; medical indications for bariatric surgery; requirements to maximize success; and a side-by-side pictorial risk/benefit comparison of the two major treatment options. For this multicenter study, individual sites made additional minor modifications to the PDA to accurately reflect state-specific insurance requirements, and program name/contact information.

#### 2.2.2. Patient/Family Survey

This 13-question survey was completed by patients/families at the end of the study encounter (Appendix A). Five of six sites set up the survey on a computer and participant responses were electronically submitted to the lead site via SurveyMonkey®. One site printed out the survey and had participants complete it by hand because of logistical constraints (i.e., computer availability, unreliable WiFi). Paper survey responses from this program were faxed to the lead site.

The first two survey items assessed parental consent and adolescent assent for study participation. Participants selected the clinic location they were seen in, but no other identifying or demographic information was collected. Survey items 5–7 were adapted from the previously published and validated three-item CollaboRATE measure of patient-reported SDM [31,32]. This measure assesses a recipient’s perception of being informed about and included in decision-making, while minimizing recall bias because of real-time delivery. CollaboRATE effectively discriminates between levels of SDM (i.e., the score increases as more dimensions of SDM are included) [32], and demonstrated utility in diverse clinical settings [33]. The remaining survey items evaluated attributes that were demonstrated to improve the quality of both the decision-making process and the decision itself [34]. These included adapted items from the validated Decisional Conflict Scale [35] and assessed self-reported knowledge and understanding of risks/benefits of each treatment option, values clarification, decisional conflict, and self-efficacy in choosing a treatment.

#### 2.2.3. Provider Feedback Questionnaire

A seven-item questionnaire was developed for this study to assess feasibility and acceptability of using the PDA in the pediatric weight management clinical setting (Appendix A). The questionnaire was administered to the provider who completed the majority of the study visits from each participating site. Open-ended questions explored facilitators and barriers to using the PDA, elicited suggestions for how to improve the PDA for future use, and asked whether the provider planned to continue to use the PDA.

### 2.3. Analysis

Responses to patient/family survey questions were assessed using frequencies, means, and standard deviations. Data completeness and the pattern of missing data were evaluated by looking at overall response rates by question and individual-level responses to each question. Qualitative responses to the provider feedback questionnaire were collated by question, grouped by common ideas, and counted.

## 3. Results

The PDA was used with 31 families (3–7 per site) over the data collection period. All 31 families provided parental consent and adolescent assent for study participation. Aggregated responses to each survey question are shown in Table 1. Overall, after use of the PDA, patient/family perceptions of provider effort to enhance their understanding, and to listen to and incorporate their preferences and values into the decision-making process were nearly maximal (8.6–8.8 out of a possible 9). The vast majority (96%) of patients/families also reported having knowledge of each option’s benefits and risks and were clear how these corresponded with their family’s priorities. A slightly lower percentage of respondents (93%) agreed that they had enough support and advice to make a choice, and fewer (89%) felt sure about what the best choice was.

### 3.1. Missing Data

There were three participants, all from the same site, who provided consent, assent, clinic location, and acknowledged being shown the PDA, but who either did not complete the rest of the survey, or whose responses to the last 10 survey questions were not properly saved and submitted. This site contributed a total of seven participants. These three individuals accounted for all but one missing data point, and are reflected in the response rates in Table 1. Pairwise deletion was used to handle the missing data (i.e., all available data from each case were used).

### 3.2. Patient/Family Survey Responses

### 3.3. Feasibility and Acceptability among Medical Providers

The provider who conducted the majority of the study visits from each of the six sites completed the provider feedback questionnaire (*n* = 6) (Appendix A). Regarding the PDA itself (i.e., content and structure), providers liked that the aid concisely presented the indications, risks, and benefits of intensive lifestyle and bariatric surgery. They thought the images of the two recommended bariatric surgical procedures were effective, and they liked that the aid contained built-in “talking points” to begin a conversation about the treatment options. Some providers preferred that the PDA include additional treatment options, like weight loss medications and medically supervised diets. Moreover, some providers expressed that the pictorial representation of risks and benefits with filled in colored dots was confusing, and that this section needed to be more closely aligned with the latest published literature. In terms of using the PDA with patients/families, providers commented that the aid facilitated an organized discussion of medical vs. surgical management with families, and allowed the provider to come “alongside the family… in the spirit of motivational interviewing”. Barriers to using the PDA with patients/families in the clinic included additional time for some but not all providers, and its lack of availability in Spanish. One provider also noted that use of the aid occasionally prompted questions about comorbidities the adolescent did not have, because they were mentioned in the PDA. Specific improvements recommended by providers included updating the aid to match the 2018 American Society for Metabolic and Bariatric Surgery pediatric guidelines [10] and latest outcomes data, including all treatment options that are available at each site, enlarging the surgical procedure graphics, and translating into Spanish. Finally, five of six sites planned on continuing to use the PDA in the future (three as is and two if revisions were made), and one of 6 reported probable future use.

## 4. Discussion

In this study, we presented a pilot application of a PDA to promote SDM for the treatment of severe obesity among adolescents across six geographically diverse pediatric weight management programs in the US. After a single use of the decision aid by the medical provider to guide a discussion of two major treatment options for severe obesity, patients/families perceived strong provider effort to embrace key principles of SDM. Predictably, not all families felt that they had enough support and advice after a single encounter to make a treatment decision or feel confident about that choice. Providers largely found the PDA to be feasible and acceptable for use in routine clinical weight management encounters, and the majority planned to use the PDA in the future. However, most made recommendations for improvements, including presentation of a broader range of available treatment options, increased clarity of evidence-based risks vs. benefits, and expanded use to non-English-speaking families. Next steps include the refinement of the PDA based on patient/family and medical provider responses, followed by an intervention trial among a larger sample of participants with pre- and post-assessment of personal determinants of decision-making (e.g., knowledge, self-efficacy) and measurement of key downstream clinical process outcomes (e.g., referrals made to bariatric surgery).

Limitations of this study include its assessment of patients/families at a single time point. This design met the primary aim of the study, but precludes any claims about changes in patient/family knowledge, beliefs, attitudes, or self-efficacy as a specific result of the decision aid intervention. Our adaptation of previously validated measures of SDM to create the family survey could influence those measures’ psychometric properties, which could affect precision and accuracy of the responses. Adolescents and their parent/caregiver were encouraged to complete the survey together, which does not allow for evaluation of the responses of either group alone or an assessment of the agreement/discordance between their responses. Participants’ baseline level of readiness to discuss the presented information was not determined, nor was the novelty of the information (e.g., was this the first or fifth time the family discussed available treatment options for severe obesity). These variables could impact outcomes of knowledge, understanding, and confidence in the decision-making process. Because this study used convenience sampling (i.e., not all eligible patients were systematically approached for participation), selection bias was introduced by the medical providers. Selection bias may also have been introduced by participants, since we do not know if there were systematic differences between those who decided to participate and those who declined. We cannot rule out that social desirability influenced patient/family survey responses. Future studies will collect demographic information (e.g., sex, race/ethnicity, socioeconomic status) to better characterize the overall sample and allow for the exploration of differences by each variable. Finally, patients were not randomized to PDA + typical clinic visit vs. typical clinic visit alone. Thus, we are unable to determine the effect of the PDA itself. Future studies will include a paired statistical analysis of outcomes (with and without the PDA for each provider) to more effectively address this question.

SDM is used across a wide variety of clinical contexts to improve the quality of decision-making in accordance with individuals’ beliefs and values. However, it remains underutilized and understudied in the field of childhood obesity [36]. A 2017 Cochrane review of 105 randomized controlled trials with decision aid interventions for disease screening or treatment included no trials that targeted children or adolescents with obesity [18]. One study by Arterburn et al. compared a video-based decision aid intervention versus a control educational booklet for bariatric surgery in adults, and demonstrated greater improvements in knowledge, decisional conflict, and more realistic postoperative outcome expectancies in the SDM group [37].

Notably, the majority of adolescents receiving bariatric surgery to date have been non-Hispanic white and female [8,25,26], despite severe obesity affecting adolescent males and females nearly equally, and disproportionately affecting minority youth [1,38]. It is not known what factors, including those that may be provider- or patient-related, affect the discussion and selection of medical and surgical treatment options in severely obese youth overall, by race/ethnicity, or by sex. Using a decision aid may increase the level of SDM, reduce bias, and facilitate equipoise in the presentation of treatment options for severely obese youth. To date, baseline rates of SDM and potential influencing factors have not been not explored among severely obese youth seeking treatment in weight management programs. By comparison, among children with attention deficit/hyperactivity disorder at treatment planning encounters in the primary care setting, the overall level of SDM behavior was observed to be low [24]. Moreover, in that study, more SDM was observed during encounters involving families with white vs. non-white children, private vs. public health insurance coverage, and higher maternal education level. It is not known whether similar disparities in SDM implementation exist when discussing treatment options available to severely obese youth, and further research is critical in this area.

The rising prevalence of severe obesity among youth poses an ongoing public health threat. Tools that promote SDM for available treatment options may not only result in better matching of treatment choice with families’ values, which could enhance adherence, but may help close the significant gap between adolescents who are medically eligible for advanced therapies and those who actually receive them.

## 5. Conclusions

The PDA used in this pilot study to compare intensive lifestyle to bariatric surgery plus lifestyle for the treatment of adolescent severe obesity was largely found to be feasible and acceptable for use in geographically diverse pediatric weight management centers in the US. Families perceived that use of the PDA at a single time point promoted central tenets of SDM. This pilot will guide a more rigorous study to determine the PDA’s effectiveness to support decision-making among adolescents with severe obesity.

## Figures and Tables

**Figure 1 ijerph-16-01776-f001:**
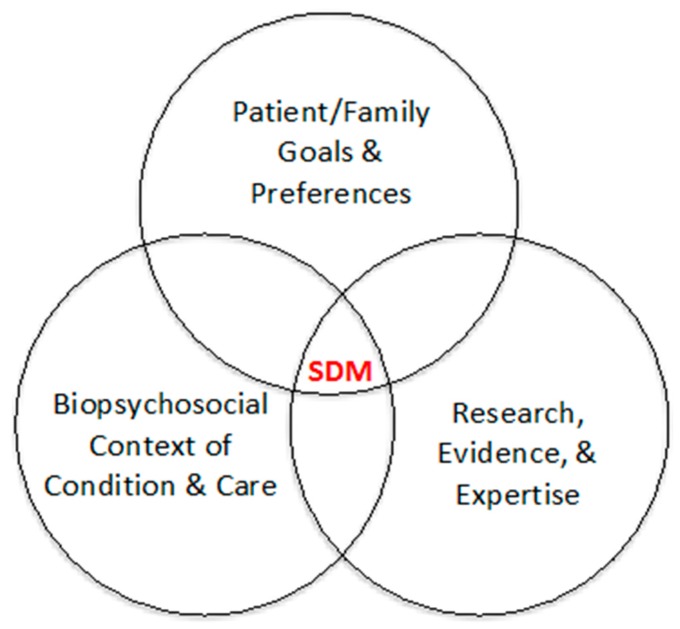
Core concepts of shared decision-making. Adapted from Cincinnati Children’s James M. Anderson Center for Health Systems Excellence definition of evidence-based and shared decision-making [20].

**Figure 2 ijerph-16-01776-f002:**
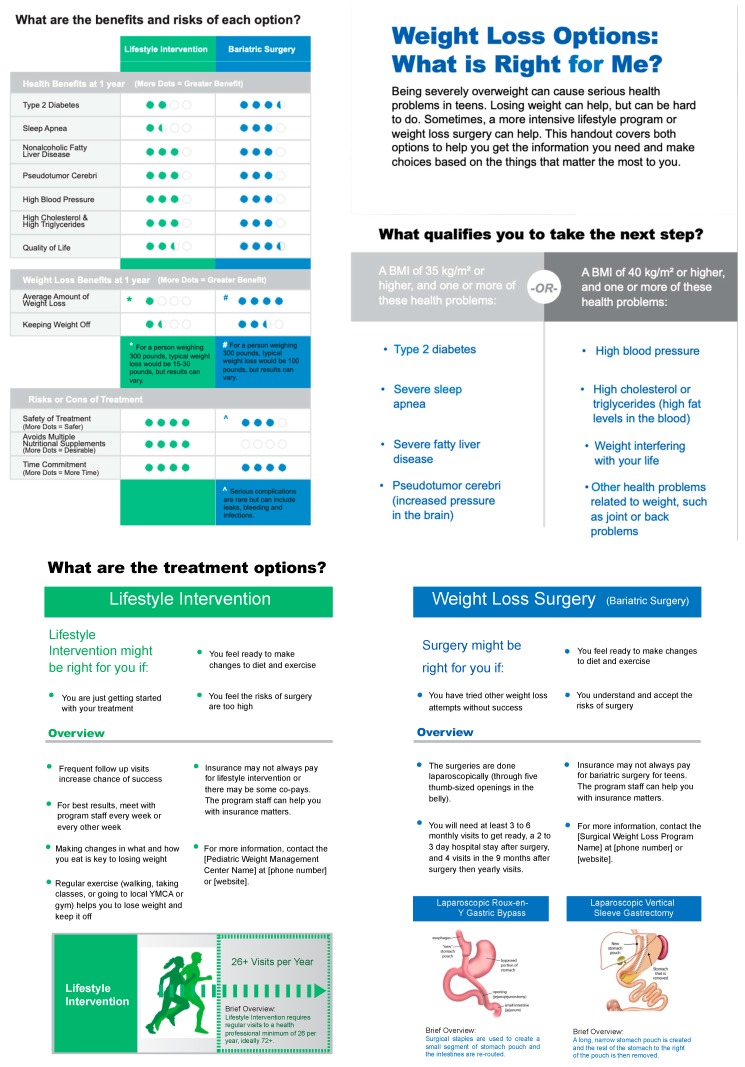
Patient decision aid.

**Table 1 ijerph-16-01776-t001:** Patient/family survey responses (*N* = 31).

Survey Question	Response Rate	Percentage “yes” or mean (SD) for questions on a 0–9 Likert scale ^1^
3. Did your clinician show you the shared decision-making tool during your visit?	31/31	100%
5. How much effort was made to help you understand your (child’s) health issues?	28/31	8.6 (0.7)
6. How much effort was made to listen to the things that matter most to you about your (child’s) health issues?	28/31	8.8 (0.4)
7. How much effort was made to include what matters most to you in choosing what to do next?	28/31	8.7 (0.5)
8. Did you discuss intensive lifestyle changes to treat your (child’s) weight?	27/31	100%
9. Did you discuss bariatric surgery (surgical weight loss) to treat your (child’s) weight?	28/31	100%
10. Do you know the benefits and risks of each option?	28/31	96%
11. Are you clear about which benefits and risks matter most to you and your child?	28/31	96%
12. Do you have enough support and advice to make a choice?	28/31	93%
13. Do you feel sure about the best choice for you (your child)?	28/31	89%

^1^ The Likert scale ranged from 0 (no effort at all) to 9 (every effort was made); SD, standard deviation.

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
