# Peer review of "Pilot Testing of a Patient Decision Aid for Adolescents with Severe Obesity in US Pediatric Weight Management Programs within the COMPASS Network"

_ijerph, 2019, doi:10.3390/ijerph16101776_

Round 1
Reviewer 1 Report
This manuscript, describing testing of a shared decision making tool newly developed to present two treatment options for severe obesity in adolescents and teenagers, provides a unique contribution to the area of patient-centered approaches to decision making and sets the stage for further study.
Some specific comments/suggestion for the authors:
- Please consider rephrasing the title to include the type of evaluation that was made (e.g., pilot testing or feasibility and acceptability of an SDM tool).
- Please elaborate on the nature of a routine clinical weight management visit.
- Elaborate on how the SDM tool was introduced and discussed within the visit. Was it incorporated at the same point in the visit across all sites or at the discretion of the provider?
- It would be helpful to relocate the explanation as to why only two forms of treatment were included in the SDM to the methods section describing the tool.
Please clarify the training procedures utilized to train the providers at the individual sites and ensure fidelity in terms of delivery of content across sites.
Author Response
REVIEWER 1:
1. Please consider rephrasing the title to include the type of evaluation that was made (e.g., pilot testing or feasibility and acceptability of an SDM tool).
Thank for you for this suggestion. The title has been changed to:
“Pilot Testing of a Patient Decision Aid for Adolescents with Severe Obesity in Pediatric Weight Management Programs within the COMPASS Network” (Note: the term “shared decision making tool” has been replaced with “patient decision aid” (PDA) throughout the manuscript to align with the International Patient Decision Aid Standards (IPDAS) Collaboration recommended nomenclature.)
2. Please elaborate on the nature of a routine clinical weight management visit.
Additional detail has been added to the Methods section (lines 173-177) to describe the components of a routine clinical weight management visit common across all sites. “Core elements of a routine visit across sites included a clinical interview (review of the medical/family/social history and medications, exploration of possible genetic/metabolic factors, and evaluation of lifestyle targets: nutrition, physical activity, and sedentary behaviors); objective assessment of weight status/trajectory using BMI; physical exam; screening for obesity comorbidities with labs and other diagnostic tests (e.g. sleep study); and treatment plan.”
3. Elaborate on how the SDM tool was introduced and discussed within the visit. Was it incorporated at the same point in the visit across all sites or at the discretion of the provider?
Detail has been added to the Methods section (lines 177-181) to elaborate on how and when the tool was introduced and discussed within the visit. “The PDA was introduced to facilitate the treatment component of the discussion, and providers led families through each section of the decision aid. Providers had discretion as to when during the visit to introduce the PDA, which varied based on practical considerations including whether it was a new or returning patient, and whether intensive lifestyle and bariatric surgery had been previously discussed.”
4. It would be helpful to relocate the explanation as to why only two forms of treatment were included in the SDM to the methods section describing the tool.
The acknowledgement and rationale for including only two forms of treatment in the tool has been expanded and added to the Methods section, under subheading 2.2.1 Patient Decision Aid (lines 228-241). “For this study, other adjunctive treatment options (e.g. medically-supervised diets, anti-obesity medications) were not included in the PDA because they a) were not uniformly offered by every program and b) have not been universally adopted as standard of care [27,28]. Expertise and resources to deliver specialized diets varied by site, and state-specific medical boards differ in their regulation of anti-obesity medications. For example, in Ohio, where the PDA was developed, the state medical board prohibits off-label use of the anorexigenic agent phentermine (including in adolescents under age 17) [29]. In other programs, phentermine is commonly prescribed [30]. In programs where treatment options other than intensive lifestyle and bariatric surgery were offered, the site PI had discretion to modify the PDA to incorporate alternative therapies. One of the six participating sites chose to make this type of modification, and added medications and special diets under the intensive lifestyle section of the PDA. Others verbally discussed available treatment options that did not appear on the PDA. This flexibility was deemed reasonable given the primary study aim of feasibility and acceptability, and helped to ensure that the full range of treatment options were presented to families.”
5. Please clarify the training procedures utilized to train the providers at the individual sites and ensure fidelity in terms of delivery of content across sites.
A description of the training for providers to improve standardization and fidelity across sites has been added to the Methods section (lines 162-165). “This study used a common protocol to help standardize delivery of the PDA across the six sites. To improve fidelity of protocol implementation, the COMPASS group trained participating providers through monthly phone calls prior to and throughout the data collection period. Topics included inclusion criteria, recruitment procedures, SDM content and delivery, and data collection.”
Reviewer 2 Report
This paper reports the results of a multi-site pre-post pilot study of the acceptability of a decision-support tool used to help 31 children/caregivers to select between two options for treating severe pediatric obesity.
Introduction:
The authors do not provide a convincing argument that this is a shared-decision-making tool – there is no discussion of provider preferences and provider attempts, within a deliberative model of care, to use persuasive communication to foster patient acceptance of treatment that best aligns with evidence-based guidelines, while respecting patient autonomy.
Methods:
There is no discussion of fidelity training and auditing to standardize the manner in which the provider used the tool. Moreover, fidelity across site in terms of training (and auditing) of the data collectors is missing.
There are no data on caregiver/child health literacy. Further, there is no evidence of a rigorous iterative design of the graphics, nor comprehension testing. In fact, providers said they found the support tool confusing, so there is a high likelihood that the materials were not understood by the children/caregivers. Without rigorous design, testing (preferences and comprehension) and refinement of health visualizations, these tools may not act as a communication or decision-making aid. For example, using pictures of internal organs (e.g. the stomach) without situating the organ inside the human body has been found to be too abstract for populations with low health literacy. The authors may find the following references helpful as they provide rigorous protocols for iterative design of health visualizations and comprehension testing.
Arcia A, Suero-Tejeda N, Bales ME, Merrill JA, Yoon S, Woollen J, et al. Sometimes more is more: Iterative participatory design of infographics for engagement of community members with varying levels of health literacy. J Am Med Inform Assoc. 2016;23(1):174-83.
Tao D, Yuan J, Qu X. Presenting self-monitoring test results for consumers: The effects of graphical formats and age. J Am Med Inform Assoc.. 2018:ocy046-ocy.
Brugger C. Public information symbols: A comparison of ISO testing procedures. Visual information for everyday use: Design and research perspectives. 1999;305–13.
Supplemental material were not available to review due to a bad link. Notwithstanding that, there is little evidence to support that the type of qualitative data generated would allow for content analysis and theme identification. It is much more likely that these were open-ended responses that were counted (qualitative themes are not “counted”).
The study was limited by the use of a tool that did not include all treatment options or that reflected current guidelines.
Comparison of length of visits when the tool was used and not used would have provided important data.
It is not clear how many children/family were approached who declined to participate and how they might have been different from those who chose to enroll.
Was an incentive offered?
The validity and reliability of surveys and questionnaires has not have established.
Discussion:
In addition to the many limitations the authors already note in the discussion section, they need to add that:
selection bias was introduced by the investigators in that not all eligible patients were systematically approached for study participation
selection bias was introduced by the subjects in that only those most motivated agreed to participate
social desirability offers an alternative explanation for survey responses
use of surveys without established reliability and validity may have influenced the precision and accuracy of responses
Author Response
REVIEWER 2:
Introduction:
1. The authors do not provide a convincing argument that this is a shared-decision-making tool – there is no discussion of provider preferences and provider attempts, within a deliberative model of care, to use persuasive communication to foster patient acceptance of treatment that best aligns with evidence-based guidelines, while respecting patient autonomy.
Thank you for your comment. Providers involved in the study agreed that the tool was to be used to augment their decision-making discussion with patients and families and thus lead to a conversation and an informed, shared decision. Providers at the lead site were extensively involved in the creation of the decision aid, which reflected a review of the available evidence-based guidelines, and their preferences about how to present the two major treatment options.
In addition, the only adolescents approached for study participation based on inclusion criteria were those who met medical criteria for bariatric surgery, as determined by the provider.
Of note, the term “shared decision making tool” has been replaced with “patient decision aid” (PDA) throughout the manuscript to align with the International Patient Decision Aid Standards (IPDAS) Collaboration recommended nomenclature. PDAs are one tool to promote shared decision making. IPDAS recommends that four components are required to meet the definition of a patient decision aid: 1) Explicit description of the decision, 2) Description of the health problem, 3) Information on options and their benefits, harms and consequences, and 4) Values clarification (implicit or explicit) (e.g. statements including “what is right for me?”).1 These four components are represented in the patient decision aid used for this study.
Reference:
1. Hoffman, A.S.; Sepucha, K.R.; Abhyankar, P.; Sheridan, S.; Bekker, H.; LeBlanc, A.; Levin, C.; Ropka, M.; Shaffer, V.; Stacey, D.; et al. Explanation and elaboration of the Standards for UNiversal reporting of patient Decision Aid Evaluations (SUNDAE) guidelines: examples of reporting SUNDAE items from patient decision aid evaluation literature. BMJ Qual Saf 2018, 27, 389–412.
Methods:
2. There is no discussion of fidelity training and auditing to standardize the manner in which the provider used the tool. Moreover, fidelity across site in terms of training (and auditing) of the data collectors is missing.
A description of the training for providers to improve standardization and fidelity across sites, has been added to the Methods section (lines 162-165). “This study used a common protocol to help standardize delivery of the PDA across the six sites. To improve fidelity of protocol implementation, the COMPASS group trained participating providers through monthly phone calls prior to and throughout the data collection period. Topics included inclusion criteria, recruitment procedures, SDM content and delivery, and data collection.”
Additional detail about when and how the decision aid was introduced by providers, has also been added to the Methods (lines 177-181). “The PDA was introduced to facilitate the treatment component of the discussion, and providers led families through each section of the decision aid. Providers had discretion as to when during the visit to introduce the PDA, which varied based on practical considerations including whether it was a new or returning patient, and whether intensive lifestyle and bariatric surgery had been previously discussed.”
We did not audit data collectors in this pilot study, but will include this step to more rigorously assess fidelity in future protocols.
3. There are no data on caregiver/child health literacy.
We did not collect caregiver/child health literacy data. To broadly maximize comprehension, the decision aid was written at an 8th grade level (Flesch-Kincaid Grade Level Index). This has been added to the Methods (line 242). We acknowledge that there are several other factors outside of a readability level that contribute to comprehension.
4. Further, there is no evidence of a rigorous iterative design of the graphics, nor comprehension testing. In fact, providers said they found the support tool confusing, so there is a high likelihood that the materials were not understood by the children/caregivers. Without rigorous design, testing (preferences and comprehension) and refinement of health visualizations, these tools may not act as a communication or decision-making aid. For example, using pictures of internal organs (e.g. the stomach) without situating the organ inside the human body has been found to be too abstract for populations with low health literacy. The authors may find the following references helpful as they provide rigorous protocols for iterative design of health visualizations and comprehension testing.
Arcia A, Suero-Tejeda N, Bales ME, Merrill JA, Yoon S, Woollen J, et al. Sometimes more is more: Iterative participatory design of infographics for engagement of community members with varying levels of health literacy. J Am Med Inform Assoc. 2016;23(1):174-83.
Tao D, Yuan J, Qu X. Presenting self-monitoring test results for consumers: The effects of graphical formats and age. J Am Med Inform Assoc.. 2018:ocy046-ocy.
Brugger C. Public information symbols: A comparison of ISO testing procedures. Visual information for everyday use: Design and research perspectives. 1999;305–13.
Development of the decision aid at Cincinnati Children’s followed IPDAS recommendations for iterative pre-testing: alpha testing with patients to check comprehensibility/usability and with providers to check acceptability/usability (both through survey feedback), followed by beta field-testing with patients and clinicians.1
The graphics within the decision aid were created with the input of a design expert within Cincinnati’s Anderson Center for Health Systems Excellence Shared Decision Making Collaborative. Thank you for providing the two references which highlight approaches to further strengthen the comprehensibility of our decision aid prior to broader effectiveness testing. As recommended in the Tao et al. article and by others,2 we avoided using raw quantitative data (opting for the preferred graphical representation: more vs. fewer dots, like in other commonly encountered rating systems). Tao et al. and Arcia et al. additionally highlight color as a technique to improve understanding, usefulness, and user confidence. Color was used in our decision aid to differentiate between the two treatment options.
Reference:
1. Coulter, A.; Stilwell, D.; Kryworuchko, J.; Mullen, P.D.; Ng, C.J.; van der Weijden, T. A systematic development process for patient decision aids. BMC Med Inform Decis Mak 2013, 13 Suppl 2, S2.
2. Shoemaker, S.J.; Wolf, M.S.; Brach, C. Development of the Patient Education Materials Assessment Tool (PEMAT): a new measure of understandability and actionability for print and audiovisual patient information. Patient Educ Couns 2014, 96, 395–403.
5. Supplemental material were not available to review due to a bad link.
Notwithstanding that, there is little evidence to support that the type of qualitative data generated would allow for content analysis and theme identification. It is much more likely that these were open-ended responses that were counted (qualitative themes are not “counted”).
Thank you for mentioning the nonfunctional link. This has been brought to the attention of the editorial staff.
Regarding the qualitative data from providers, common ideas expressed by providers were grouped, counted and summarized. The wording describing the analysis of this data has been modified in Methods subsection 2.3 to (lines 302-304): “Qualitative responses to the provider feedback questionnaire were collated by question, grouped by common ideas, and counted.”
6. The study was limited by the use of a tool that did not include all treatment options or that reflected current guidelines.
Anti-obesity treatment options
The rationale for including only two forms of treatment in the tool has been expanded and added to the Methods section, under subheading 2.2.1 Patient Decision Aid (lines 228-241) as follows: “For this study, other adjunctive treatment options (e.g. medically-supervised diets, anti-obesity medications) were not included in the PDA because they a) were not uniformly offered by every program and b) have not been universally adopted as standard of care [27,28]. Expertise and resources to deliver specialized diets varied by site, and state-specific medical boards differ in their regulation of anti-obesity medications. For example, in Ohio, where the PDA was developed, the state medical board prohibits off-label use of the anorexigenic agent phentermine (including in adolescents under age 17) [29]. In other programs, phentermine is commonly prescribed [30]. In programs where treatment options other than intensive lifestyle and bariatric surgery were offered, the site PI had discretion to modify the PDA to incorporate alternative therapies. One of the six participating sites chose to make this type of modification, and added medications and special diets under the intensive lifestyle section of the PDA. Others verbally discussed available treatment options that did not appear on the PDA. This flexibility was deemed reasonable given the primary study aim of feasibility and acceptability, and helped to ensure that the full range of treatment options were presented to families.”
Guidelines
The decision aid reflected current clinical guidelines at the time of its creation. Not unexpectedly since that time there have been additional data that will be used to refine the aid prior to its next phase of study. This new data does not result in any change in direction of the risk/benefit comparison between lifestyle and bariatric surgery (e.g. bariatric surgery is still superior to lifestyle in treating type 2 diabetes). Additionally, the most recent American Society for Metabolic and Bariatric Surgery pediatric guidelines were not published until 2018, after data collection for this study was completed.
7. Comparison of length of visits when the tool was used and not used would have provided important data.
This is a valid point. However, the open-ended question to providers that asked “Were there any barriers to using the SDM tool with patients/families in clinic?” could reasonably be expected to capture time/prolongation of visits as a practical impediment to using the decision aid. This barrier was noted by a minority of providers.
8. It is not clear how many children/family were approached who declined to participate and how they might have been different from those who chose to enroll.
We do not have this data, and plan to collect this in the next phase of study.
9. Was an incentive offered?
No incentive was offered to families or medical providers. This point has been explicitly added to Methods (lines 170-171).
10. The validity and reliability of surveys and questionnaires has not have established.
Relevant detail about the two validated instruments that informed the patient/family survey have been added to the Methods (lines 282-291): “Survey items 5-7 were adapted from the previously published and validated 3-item CollaboRATE measure of patient-reported shared decision making [30,31]. This measure assesses a recipient’s perception of being informed about and included in decision making, while minimizing recall bias because of real-time delivery. CollaboRATE effectively discriminates between levels of SDM (i.e. the measure’s score increases as more dimensions of SDM are included) [31], and has demonstrated utility in diverse clinical settings [32]. The remaining survey items evaluated attributes that have been demonstrated to improve the quality of both the decision making process and the decision itself [33]. These included adapted items from the validated Decisional Conflict Scale [34] and assessed self-reported knowledge and understanding of risks/benefits of each treatment option, values clarification, decisional conflict, and self-efficacy in choosing a treatment.”
The questionnaire for providers was developed specifically for this study.
Discussion:
11. In addition to the many limitations the authors already note in the discussion section, they need to add that:
· Selection bias was introduced by the investigators in that not all eligible patients were systematically approached for study participation.
· Selection bias was introduced by the subjects in that only those most motivated agreed to participate.
· Social desirability offers an alternative explanation for survey responses
Thank you for these suggestions. Acknowledgement of selection bias and possible social desirability has been added to the Discussion (lines 451-455).
12. Use of surveys without established reliability and validity may have influenced the precision and accuracy of responses.
While the patient/family survey was based on valid instruments, we understand that any modification of these measures could alter their psychometric properties. This limitation has been added to the Discussion (lines 442-445).
Reviewer 3 Report
Title: The title should be more specific, such as including “Pilot study” in the title
Introduction:
The objectives of the study are too vague. It would be better to quantify in order to claim as a success study.
Materials and Methods:
In study Design and Procedures, the author indicated that six programs participated in the study (line 79). However, there are 12 settings listed in the patient/family survey. Why is the discrepancy?
The authors need to describe more in details on how they enrolled patients.
With the prevalence of 8.5% of 12-19 year old severe obesity, the final sample of the study seem to be pretty low (only 31 patients collected in 10 months of the study period within 6 locations, 27 responses left for analyze). Have the authors did the power calculation prior to the study to determine how many should they need to collect?
Please explain more on how paper copy of surveys at different locations storing at the Cincinnati Children’s site.
Results:
The Shared Decision Making Tool has been finalized in April 2015, but the study did not take place until December 2016. What happened in between? Also, although the tool has been field tested iteratively, many providers still find some of the contents confusing. In that case, the validity of the tool seem needs to be improved.
How many providers participated in the Provider Feedback Questionnaire?
Discussion:
Since the authors lacking no providing any other results/information except for the patient/family survey, without the comparison group or patient/family characteristics, I would be hesitated to draw the conclusion that “this SDM tool showed promising utility to support patient/family decision making for severe obesity treatment.”
The limitations the authors listed were mostly major points, such as only assess patients/families at a single time point, no patients/families demographics, and no comparison groups. These are still essential factors to be considered in a pilot study.
Reference: link in reference 9 did not provide sufficient information related to the content. Please update the link.

Author Response
REVIEWER 3:
Title:
1. The title should be more specific, such as including “Pilot study” in the title
Thank for you for this suggestion. The title has been changed to:
“Pilot Testing of a Patient Decision Aid for Adolescents with Severe Obesity in Pediatric Weight Management Programs within the COMPASS Network” (Note: the term “shared decision making tool” has been replaced with “patient decision aid” (PDA) throughout the manuscript to align with the International Patient Decision Aid Standards (IPDAS) Collaboration recommended nomenclature.)
Introduction:
2. The objectives of the study are too vague. It would be better to quantify in order to claim as a success study.
More specificity has been added to the study aims, including accompanying a priori hypotheses, lines 87-95 as follows: “Thus, the primary aim of this study was 1) to establish whether a PDA for the treatment of adolescent severe obesity was feasible and acceptable to implement within usual pediatric weight management care. Secondary aims were 2) to determine whether use of the PDA could promote core SDM principles at a single time point as perceived by adolescents with severe obesity and their families, and 3) to further refine the PDA using data from patient/family surveys and providers. We hypothesized that providers at 100% of participating sites would be able to incorporate the PDA into their clinical visits, that family survey data would reflect an experience inclusive of multiple dimensions of SDM, and that family and provider feedback would suggest further enhancements of the PDA to improve its usability and acceptability.”
Materials and Methods:
3. In study Design and Procedures, the author indicated that six programs participated in the study (line 79). However, there are 12 settings listed in the patient/family survey. Why is the discrepancy?
Thank you for pointing out this discrepancy. There were a total of 12 COMPASS sites that initially expressed interest in study participation. However, only six of the 12 ultimately obtained IRB approval and collected data during the study period. This detail has been added to the Methods (line 154).
4. The authors need to describe more in details on how they enrolled patients.
For this pilot study, each site had the goal of recruiting a convenience sample of 5 families who met inclusion criteria during new or follow-up clinical encounters. This detail has been added to the Methods (lines 167-168). Convenience sampling increases the risk for selection bias, a limitation that has been added to the Discussion, (lines 451-452).
5. With the prevalence of 8.5% of 12-19 year old severe obesity, the final sample of the study seem to be pretty low (only 31 patients collected in 10 months of the study period within 6 locations, 27 responses left for analyze). Have the authors did the power calculation prior to the study to determine how many should they need to collect?
The target sample size was 30 (5 families per site using a convenience sample). Thus, only a fraction of all potentially eligible 12-17 year olds with severe obesity seen were included. No formal power calculation was made because our primary aim was not structured around any specific effectiveness measure. However, we did keep in mind that a minimum of 12 individuals for pilot studies is recommended to be able to estimate mean and variability for the continuous shared decision making variables reported from the patient family survey.1 These estimates will be used to help inform the next study of the decision aid’s effectiveness.
Reference
1. Julious, S.A. Sample size of 12 per group rule of thumb for a pilot study. Pharmaceutical Statistics 2005, 4, 287–291.
6. Please explain more on how paper copy of surveys at different locations storing at the Cincinnati Children’s site.
Four out of the five site principal investigators chose to administer the patient/family survey online, for which responses were electronically stored, and accessible only to the lead site PI: RS. One site PI preferred to print out the survey and have participants complete them by hand because of logistical concerns (e.g. computer availability, reliable WiFi). All paper survey responses were faxed to RS upon completion, and receipt was confirmed by RS during the subsequent monthly COMPASS phone call.
Results:
7. The Shared Decision Making Tool has been finalized in April 2015, but the study did not take place until December 2016. What happened in between? Also, although the tool has been field tested iteratively, many providers still find some of the contents confusing. In that case, the validity of the tool seem needs to be improved.
Between April 2015 and December 2016, the original study protocol was developed by the lead site, Cincinnati Children’s, and IRB approval was obtained at that institution. Then, the study was proposed to the broader COMPASS group. The interested COMPASS sites came to group consensus about a unified protocol, decision aid contents, delivery approach, and data collection methodology, then each of the additional interested sites obtained local IRB approval prior to data collection.
Development of the decision aid at Cincinnati Children’s followed IPDAS recommendations for iterative pre-testing: alpha testing with patients to check comprehensibility/usability and with providers to check acceptability/usability (both through survey feedback), followed by beta field-testing with patients and clinicians.1 It is not unexpected that expansion of the testing to 5 additional sites resulted in additional suggestions to improve comprehension and usability. We agree that improvements will need to be made before the next phase of effectiveness testing for the decision aid.
Reference:
1.Coulter, A.; Stilwell, D.; Kryworuchko, J.; Mullen, P.D.; Ng, C.J.; van der Weijden, T. A
systematic development process for patient decision aids. BMC Med Inform Decis Mak 2013,
Suppl 2, S2.
8. How many providers participated in the Provider Feedback Questionnaire?
Six providers (1 from each site) provided responses to the questionnaire. The text has now been clarified in the Methods Lines 190-192 and Results Lines: 368-369 to “The provider who completed the majority of the SDM study visits from each of the six participating sites completed the provider feedback questionnaire (n=6) (Table S1).”
Discussion:
9. Since the authors lacking no providing any other results/information except for the patient/family survey, without the comparison group or patient/family characteristics, I would be hesitated to draw the conclusion that “this SDM tool showed promising utility to support patient/family decision making for severe obesity treatment.”
Thank you for this suggestion. We have reworded our Conclusion Lines 536-541 to better align with the aims of the study and planned next steps as follows: “The PDA used in this pilot study to compare intensive lifestyle to bariatric surgery plus lifestyle for the treatment of adolescent severe obesity was largely found to be feasible and acceptable for use in geographically diverse pediatric weight management centers in the US. Families perceived that use of the PDA at a single time point promoted shared decision making tenets. This pilot will guide a more rigorous study to determine the PDA’s effectiveness to support decision making among adolescents with severe obesity.”
10. The limitations the authors listed were mostly major points, such as only assess patients/families at a single time point, no patients/families demographics, and no comparison groups. These are still essential factors to be considered in a pilot study.
We agree that addressing the limitations noted would significantly strengthen the study. During the study design process, the decisions not to randomize, to assess at a single time point, and not to collect demographic information from participants were driven by the goal of achieving the primary study aim, which were to assess feasibility of using the SDM tool in multiple busy pediatric weight management practices, and secondary aim to assess initial patient/family perceptions of shared decision making factors. Practical considerations for achieving these goals, while minimizing participant burden, were considered.
11. Reference: link in reference 9 did not provide sufficient information related to the content. Please update the link.
Reference 9 links to the most current population data available from the US Census Bureau. This raw data was used to manually calculate the total number of males and females age 12 through 18 in the US in 2017. This was then multiplied by the most current prevalence estimate of severe obesity in adolescents (8.5%) to derive the number of severely obese adolescents, which represents the largest number of adolescents who could be eligible for bariatric surgery based on a diagnosis of severe obesity alone. Interestingly, there is no one agreed upon definition of “adolescence”. The World Health Organization and US Department of Health and Human services define it as ages 10-19, the American Academy of Pediatrics (AAP) defines it as ages 11-21, and in clinical practice many define it as ages 12-18. In thinking through your comment, we have decided to update our estimate to be consistent with the ages of adolescence as defined by the AAP = 46,313,244 total 11-21 year olds x 0.085= 3.9 million.
Reviewer 4 Report
I commend the authors for investigating SDM in discussion of potential bariatric surgery but I have a few concerns.
-While I understand that there is under-utilization of bariatric surgery for those who are eligible, I wonder if the patients recruited for this study had previously underwent more conservative treatments first (e.g. supervised diets, pharmacotherapy).
-In reference to the SDM tool, there are only two options presented and if the tool is to be refined, I would recommend to include more available treatment options.
-Additionally, the SDM tool should adhere to the IPDASI standards [Elwyn et al, Assessing the Quality of Decision Support Technologies Using the International Patient Decision Aid Standards instrument (IPDASi)]
-While the authors do acknowledge the limitations of the study design--i.e. patients were not randomized to SDM tool vs usual clinic visit without SDM tool--this is a major limitation of the study as a pre-post study would be able to show if the tool truly had a impact on SDM in comparison to usual care.
-The authors used patient-reported measures of SDM--Currently, there are no validated measures of SDM and this should be included in the limitation part of the discussion.
Author Response
REVIEWER 4
1. While I understand that there is under-utilization of bariatric surgery for those who are eligible, I wonder if the patients recruited for this study had previously underwent more conservative treatments first (e.g. supervised diets, pharmacotherapy).
All patients recruited had exposure to lifestyle-based interventions. We did not collect the specific nature/duration of prior treatment exposures. However, in general, and in accordance with the 2007 Expert Committee Guidelines for pediatric obesity1, obesity treatment is delivered with increasing stages of intensity, tailored to family preferences/motivation and the severity of obesity. The highest intensity treatment stage includes anti-obesity medications, specialized diets, and bariatric surgery. The treatment options were not uniform across sites (please also see response to comment 2 below).
Reference
1. Barlow, S.E.; Expert Committee Expert committee recommendations regarding the prevention, assessment, and treatment of child and adolescent overweight and obesity: summary report. Pediatrics 2007, 120 Suppl 4, S164-192.
2. In reference to the SDM tool, there are only two options presented and if the tool is to be refined, I would recommend to include more available treatment options.
Medically supervised diets and anti-obesity medications were not offered by every site. However, providers whose programs did offer these adjunctive treatments had discretion to discuss these, and one out of the six programs modified the decision aid to include them. Other providers whose sites offered alternative treatment options verbally discussed these while presenting the decision aid to families.
The rationale for including only two forms of treatment in the original decision aid has been expanded and added to the Methods section, under subheading 2.2.1 Patient Decision Aid (lines 228-241). Before the next phase of study, we agree that providing a decision aid that can be tailored to represent all available treatment options is a priority.
3. Additionally, the SDM tool should adhere to the IPDASI standards [Elwyn et al, Assessing the Quality of Decision Support Technologies Using the International Patient Decision Aid Standards instrument (IPDASi)]
The decision aid was designed to adhere to IPDAS standards.
IPDAS recommends that four components are required to meet the definition of a patient decision aid: 1) Explicit description of the decision, 2) Description of the health problem, 3) Information on options and their benefits, harms and consequences, and 4) Values clarification (implicit or explicit) (e.g. statements including “what is right for me?”).1 These four components are represented in the patient decision aid used for this study.
Additionally, development of the decision aid followed IPDAS recommendations for iterative pre-testing: alpha testing with patients to check comprehensibility/usability and with providers to check acceptability/usability (both through survey feedback), followed by beta field-testing with patients and clinicians.2
Of note, the term “shared decision making tool” has now been replaced with “patient decision aid” (PDA) throughout the manuscript to align with the International Patient Decision Aid Standards (IPDAS) Collaboration recommended nomenclature.
Reference:
1.Hoffman, A.S.; Sepucha, K.R.; Abhyankar, P.; Sheridan, S.; Bekker, H.; LeBlanc, A.; Levin, C.; Ropka, M.; Shaffer, V.; Stacey, D.; et al. Explanation and elaboration of the Standards for UNiversal reporting of patient Decision Aid Evaluations (SUNDAE) guidelines: examples of reporting SUNDAE items from patient decision aid evaluation literature. BMJ Qual Saf 2018, 27, 389–412.
2. Coulter, A.; Stilwell, D.; Kryworuchko, J.; Mullen, P.D.; Ng, C.J.; van der Weijden, T. A systematic development process for patient decision aids. BMC Med Inform Decis Mak 2013, Suppl 2, S2.
4. While the authors do acknowledge the limitations of the study design--i.e. patients were not randomized to SDM tool vs usual clinic visit without SDM tool--this is a major limitation of the study as a pre-post study would be able to show if the tool truly had a impact on SDM in comparison to usual care.
We agree that addressing the limitations noted would have significantly strengthened the study. During the study design process, the decisions not to randomize, to assess at a single time point, and not to collect demographic information from participants were driven by the goal of achieving the primary study objectives, which were to assess feasibility of using the SDM tool in multiple busy pediatric weight management practices, and to assess initial patient/family perceptions of shared decision making factors. Practical considerations for achieving these goals, while minimizing participant burden, were considered. We plan to address this limitation in the future effectiveness study of this decision aid.
5. The authors used patient-reported measures of SDM--Currently, there are no validated measures of SDM and this should be included in the limitation part of the discussion.
We used two previously validated shared decision making instruments to develop the patient/family survey. Additional detail has been added regarding the validity of these measures: CollaboRATE1 and the Decision Conflict Scale2. Lines 282-291
References
1. Barr, P.J.; Thompson, R.; Walsh, T.; Grande, S.W.; Ozanne, E.M.; Elwyn, G. The psychometric properties of CollaboRATE: a fast and frugal patient-reported measure of the shared decision-making process. J. Med. Internet Res. 2014, 16, e2.
2. O’Connor, A.M. Validation of a decisional conflict scale. Med Decis Making 1995, 15, 25–30.
Round 2
Reviewer 4 Report
Thank you for providing the revisions and clarifications, which has strengthened the paper.
Author Response
Comment 1: Thank you for providing the revisions and clarifications, which has strengthened the paper.
Response: We appreciate the feedback and agree that the manuscript is stronger based on the suggested revisions.